# Vaccine-elicited IL-1R signaling results in Th17 TRM-mediated immunity
Joseph P. Hoffmann [1], Akhilesh Srivastava[1], Haoran Yang [1], Naoki Iwanaga [1,2], T. Parks Remcho[1], Jenny L. Hewes[1], Rayshma Sharoff[1], Kejing Song[1], Elizabeth B. Norton[3], Jay K. Kolls [1] & Janet E. McCombs [1] ✉

Lung tissue resident memory (TRM) cells are thought to play crucial roles in lung host defense. We have recently shown that immunization with the adjuvant LTA1 (derived from the A1 domain of *E. coli* heat labile toxin) admixed with OmpX from *K. pneumoniae* can elicit antigen specific lung Th17 TRM cells that provide serotype independent immunity to members of the *Enterobacteriaceae* family. However, the upstream requirements to generate these cells are unclear. Single-cell RNA-seq showed that vaccine-elicited Th17 TRM cells expressed high levels of IL-1R1, suggesting that IL-1 family members may be critical to generate these cells. Using a combination of genetic and antibody neutralization approaches, we show that Th17 TRM cells can be generated independent of caspase-1 but are compromised when IL-1α is neutralized. Moreover IL-1α could serve as a molecular adjuvant to generate lung Th17 TRM cells independent of LTA1. Taken together, these data suggest that IL-1α plays a major role in vaccine-mediated lung Th17 TRM generation.

The HIV epidemic clearly demonstrated the critical roles CD4+ T cells play in host resistance to pneumonia. As such, development of vaccines that elicit these cells at mucosal sites of infection is of great interest. Prior work from our group has shown that complex antigen exposure, such as immunization with heat-killed *Klebsiella pneumoniae*, results in the generation of protective lung Th17 TRM cells in mice[1]. We further demonstrated a mucosal subunit vaccine consisting of the bacterial outer membrane protein OmpX from *K. pneumoniae* and the adjuvant composed of the A1 domain of *E. coli* heat labile toxin (LTA1)[2] also induces a robust enrichment of lung Th17 TRM cells that protect against pulmonary challenge with *K. pneumoniae*[3]. Given OmpX is a highly conserved bacterial protein, these elicited T cells were able to recognize other *Enterobacteriaceae* family members that also encode OmpX. Development of such broadly acting vaccines offers an advantage over traditional vaccines that only elicit humoral responses. However, the relative roles of upstream cytokines in directing the development of vaccine-elicited Th17 TRM cells from naïve T cell progenitors remain unknown.

Studies examining the effects of LTA1 in THP-1 cells found LTA1 activates the NLRP3 (NOD-, LRR- and pyrin domain-containing protein 3) inflammasome, leading to cleavage of pro-caspase-1 into caspase-1 and subsequent secretion of IL-1β[4]. Importantly, IL-1R1 signaling has been linked to the activation and expansion of Th17 cells previously[5,6]. In particular, IL-1β promotes Th17 cell differentiation in humans and mice and enhances memory T cell activation and proliferation[7-11]. A similar role for IL-1α in survival and expansion of memory CD4+ T cells has also been suggested[10]. However, whether IL-1 signaling is critical for vaccine-mediated Th17 cell generation is unknown. A greater understanding of the mechanisms through which naïve T cells develop into Th17 TRM cells upon immunization would enable development of improved mucosal vaccines.

Analysis of single-cell RNA sequencing (sc-RNA-seq) data we generated previously[3] revealed that immunization with OmpX + LTA1 via oropharyngeal aspiration induced lung CD4+ T cells expressing high levels of IL-1R1 compared to naïve T cells isolated from the spleen. Based on these sc-RNA-seq data and previous studies linking IL-1R1 signaling to TRM generation and Th17 cell expansion, we postulated that members of the IL-1 family may be critical for the generation of lung Th17 TRM cells in the context of our vaccine. Using a combination of genetic models and antibody depletion studies, we demonstrated a role for IL-1-related cytokines in Th17 TRM cell generation following immunization with OmpX + LTA1. In addition, we assessed the use of these cytokines as mucosal adjuvants in a *K. pneumoniae* vaccine.

[1]Center for Translational Research in Infection and Inflammation, Tulane University School of Medicine, New Orleans, LA, USA. [2]Department of Respiratory Medicine, Nagasaki University Hospital, Nagasaki, Japan. [3]Department of Immunology and Microbiology, Tulane University School of Medicine, New Orleans, LA, USA. ✉e-mail: jmccombs@tulane.edu

## Results

### Mucosal Immunization with OmpX + LTA1 induces IL-1 signaling in vivo

To identify upstream signaling pathways that could play a role in formation of Th17 TRM cells upon mucosal vaccination via oropharyngeal aspiration, we examined our existing sc-RNA-seq data comparing CD4+ T cells isolated from the lungs of mice immunized with OmpX+LTA1 and naïve CD4+ T cells from spleens of unimmunized mice[3]. Comparison to naïve splenic T cells was chosen due to the lack of lung resident CD4+ T cells in unvaccinated SPF mice. These groups displayed distinct clustering in Uniform Manifold Approximation and Projection (UMAP) plots,

suggesting a shift in gene expression of CD4+ T cells following immunization (Fig. 1a). Analysis of these clusters for gene expression differences indicated upregulation of *Il1r1* transcript levels in the lung resident CD4+ T cells, while there was no detectable *Il1r1* in the CD4+ T cells from spleens of naïve mice (Fig. 1b). Using flow cytometry, we also confirmed that binding of an anti-IL-1R1 antibody to CD4+ T cells was increased in cells isolated from the lungs of vaccinated mice compared to PBS treated mice, suggesting vaccination increased expression of this receptor on these cells (Fig. 1c).

To determine the impact of vaccination on inflammatory responses in vivo, C57Bl/6 mice were immunized twice, three weeks apart with

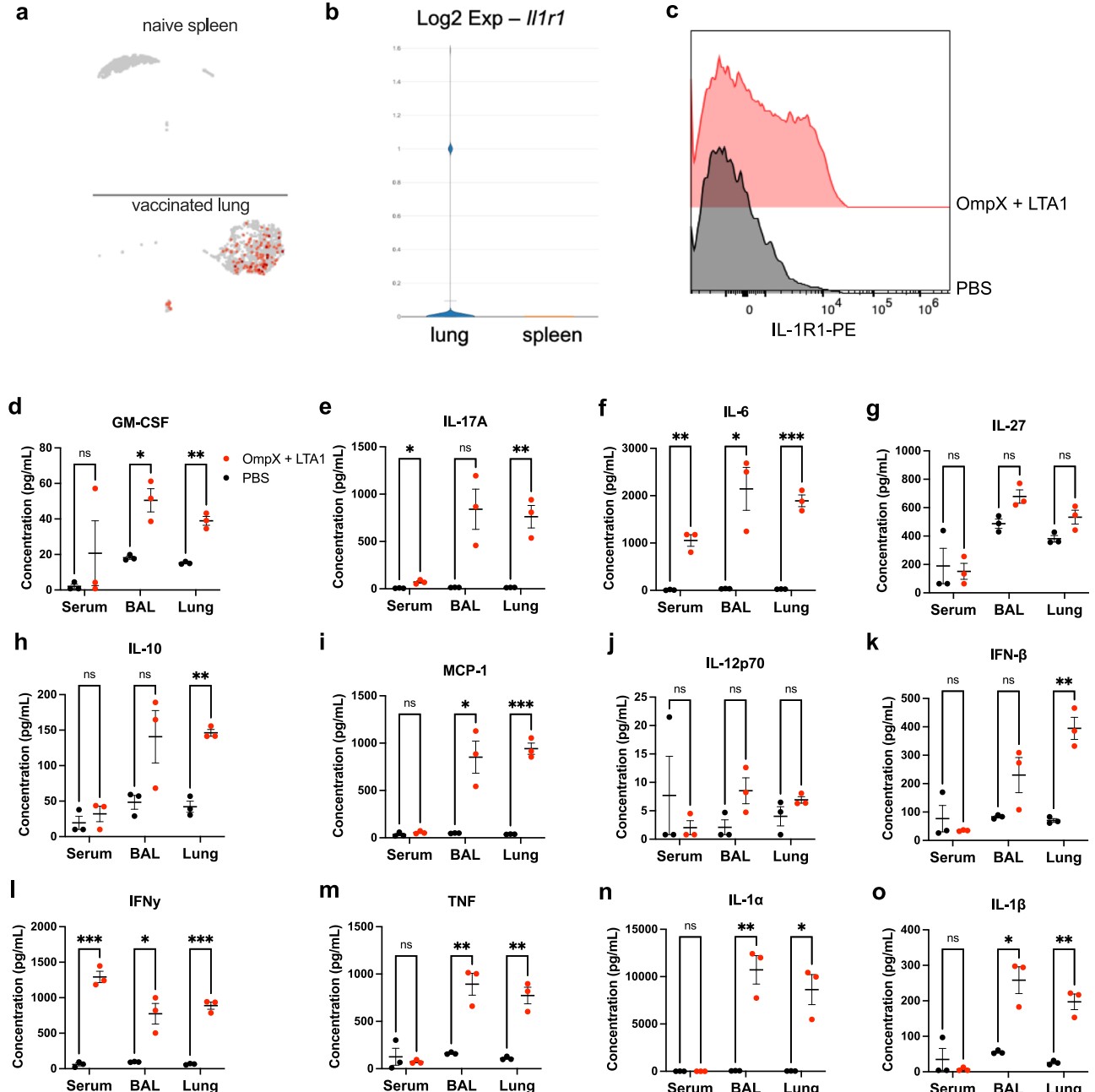

**Fig. 1 | OmpX + LTA1 immunization promotes *Il1r1* transcript expression in CD4 T cells and induces IL-1 and other proinflammatory cytokines *in vivo*.**
**a** Representative UMAP plot of *Il1r1* expression in CD4+ cells from naïve spleens and OmpX+LTA1 immunized lungs. **b** The Log2 expression levels of *Il1r1* from lung CD4+ T cells (lung) and naïve splenic CD4+ T cells (spleen) shown in **a**. **c** Representative flow cytometry histograms showing IL-1R1 on CD4+ T cells

(CD45+, CD19-, CD3+, CD4+) from single cell lung suspensions of PBS treated and OmpX+LTA1 vaccinated mice. Concentrations of (**d**) GM-CSF, (**e**) IL-17A, (**f**) IL-6, (**g**) IL-27, (**h**) IL-10, (**i**) MCP-1, (**j**) IL-12p70, (**k**) IFN-β, (**l**) IFNγ, (**m**) TNF, (**n**) IL-1α, and (**o**) IL-1β in the serum, BAL fluid, and lung homogenate of immunized and unimmunized mice. Data are presented as the mean +/− SEM, *n* = 3 mice per group. Data were analyzed using the unpaired T test.

OmpX + LTA1 or administered PBS via oropharyngeal aspiration as previously described[3]. One day following the boost, mice were euthanized and the presence of inflammatory cytokines was assessed in the serum, lung tissue homogenate, and BALF of vaccinated mice and PBS administered controls using the LegendPlex mouse inflammation panel. As expected, vaccinated mice had significantly higher expression of proinflammatory cytokines than their PBS counterparts, particularly in the lungs, as measured in the lung homogenate and BALF samples (Fig. 1d–o). For recruitment and expansion of innate immune cells, we observed greater expression of GM-CSF and MCP-1 in the lungs (Fig. 1d, i). As demonstrated previously for LTA1[3], we saw a significant increase in IL-17A in the lung homogenate, and a small but significant increase in the serum (Fig. 1e). The BALF values for IL-17A were not significant ($p = 0.054$) but demonstrated a trend that matched the other samples tested. Additionally, we found vaccination with OmpX+LTA1 led to an increase of the proinflammatory cytokines IL-6, IFNγ, and TNF in the lung homogenate and BALF, with IL-6 and IFNγ also being enhanced systemically as demonstrated by increased levels of these cytokines in the serum (Fig. 1f, l, m). No significant increases in the IL-12 family cytokines IL-12p70 and IL-27, which promote IFNγ production in T cells and inhibit Th17 cell differentiation, respectively[12], were observed (Fig. 1g, j). Interestingly, IFN-β, which negatively regulates Th17[13], as well as the anti-inflammatory cytokine IL-10, were increased in lung homogenates (Fig. 1h, k). Importantly, we found that immunized mice had significantly elevated amounts of the IL-1R1-related ligands IL-1α and IL-1β in the lungs with the amounts of IL-1α particularly striking (Fig. 1n, o). All together, these data support previously described in vitro observations[4] and suggest vaccination with OmpX+LTA1 induces an inflammatory response and expression of the IL-1R1 receptor and its IL-1 cytokine ligands in our model.

## IL-1 signaling promotes formation of vaccine-elicited mucosal Th17 cells

After verifying that our vaccine promotes the expression of IL-1 cytokines in the respiratory tract in vivo, we next investigated whether IL-1 signaling was necessary for generating enriched adaptive immune cell populations and protective immunity. To do this, we utilized Il1r1 global knockout mice (Il1r1⁻/⁻). These mice are deficient in the receptor IL-1R1, which both IL-1β and IL-1α can bind to and signal through via MyD88 and subsequently NFκB[14]. As such, Il1r1⁻/⁻ mice are unable to respond to IL-1β or IL-1α. Both wildtype and Il1r1⁻/⁻ mice were immunized with either OmpX + LTA1 or administered PBS as above to determine the effects of IL-1 signaling on vaccine induced immunity. One week following the final boost all mice were challenged with $1 \times 10^4$ CFU K. pneumoniae and euthanized 24 h after challenge to assess immune cell populations and bacterial burdens.

To evaluate adaptive immune cell populations, we quantified B cells, CD4+ T cells, and Th17 cells in the lungs of all animals using flow cytometry following the gating strategy in Supplementary Fig. 1. Examination of CD4+ T cell and B cell populations in the lungs of immunized and naïve mice revealed vaccination increased the abundance of these cells in both the wildtype and Il1r1⁻/⁻ mice compared to PBS-treated controls, though there was no difference in number of CD4+ T cells between the immunized Il1r1⁻/⁻ mice and their wildtype counterparts (Fig. 2a, b). Given these data, it appears that IL-1 signaling does not have an impact on the enrichment of B cell and CD4+ T cell populations in our vaccine model. Interestingly, though vaccinated Il1r1⁻/⁻ and wildtype mice had comparable levels of total lung CD4+ T cells, immunized Il1r1⁻/⁻ had a significantly smaller population of Th17 cells (Fig. 2c and d). Indeed, roughly half as many CD4+ T cells were IL-17A+ in the knockout compared to wildtype immunized mice and there was a trend toward reduced IL-17A median fluorescence intensity (MFI) in knockout versus the wildtype mice (Fig. 2e). Of note, PBS treated wildtype and Il1r1⁻/⁻ mice had similar populations of B cells, CD4+ T cell, and Th17 cells in the lungs (Fig. 2a–d), suggesting knockout of IL-1 signaling does not impact these populations in naïve mice. To confirm observed changes in immune cell populations were vaccine driven and not due to challenge, we evaluated CD4+ T cells, B cells, and Th17 cells in unchallenged immunized mice. Interestingly, there were higher levels of

CD4+ T cells in wildtype versus Il1r1⁻/⁻ mice upon vaccination (Supplementary Fig. 2a). While a similar trend was observed in challenged mice, these differences were not significant (Fig. 2a). Similar to our findings post-challenge, there were no differences in B cells between groups following vaccination (Supplementary Fig. 2b). In addition, Th17 cells were increased upon vaccination in both Il1r1⁻/⁻ and wildtype mice when compared to PBS treated mice, with a higher number of Th17 cells in wildtype compared to Il1r1⁻/⁻ vaccinated mice (Supplementary Fig. 2c). We also observed a greater percentage of CD4+ T cells that were IL-17A+ in wildtype versus Il1r1⁻/⁻ vaccinated mice (Supplementary Fig. 2d). Interestingly, only the wildtype vaccination group generated OmpX-specific IL-17A-secreting T cells when compared with the PBS administered group, as evidenced by ELISpot (Supplementary Fig. 2e). We previously demonstrated that the majority (~90%) of vaccine-elicited Th17 cells in the lungs were tissue resident[3], suggesting IL-1R1 signaling may play a role in Th17 TRM generation. Indeed, characterization of TRM cells using the cell surface markers CD44 and CD69 revealed that while TRMs were increased upon vaccination of both wildtype and Il1r1⁻/⁻ mice, knockout of IL-1R1 resulted in a decrease in this population post-vaccination when compared to wildtype mice (Supplementary Fig. 2f, g).

We next evaluated presence of K. pneumoniae specific IgA in lung homogenates. Both immunized wildtype and Il1r1⁻/⁻ mice had levels of lung IgA that were significantly higher compared to the respective PBS administered groups, though no differences were seen between vaccinated groups. (Fig. 2f, g). Thus, observed immunological differences between the two immunized groups appear to be restricted to Th17 cell populations. Since we have previously demonstrated that Th17 cells are necessary for vaccine mediated protection against pulmonary challenge with K. pneumoniae[3], we next assessed bacterial burdens in the lungs and spleens from each group 24 h post-challenge with K. pneumoniae. Surprisingly, though the immunized Il1r1⁻/⁻ mice had significantly fewer Th17 cells than the wildtype mice, both groups were protected from challenge as demonstrated by bacterial burdens in the lungs and spleens (Fig. 2h, i). There was no significant difference between the immunized groups and both groups had significantly reduced burdens from their PBS administered counterparts. Taken together, it appears that IL-1 signaling is important but not the only requirement for the expansion of vaccine-induced Th17 cells. While mice without IL-1 signaling had reduced populations of Th17 cells, these cell numbers appeared to be sufficient in conferring protection in our vaccination and challenge model.

## Cleaved IL-1β is dispensable for Th17 cell expansion and vaccine mediated protection

To further investigate how IL-1 signaling influences the expansion of Th17 cells in our vaccination model, we utilized mice deficient in caspase-1 (Casp1⁻/⁻)[15]. Caspase-1, which is cleaved and activated by the inflammasome complex, proteolytically cleaves cytosolic pro-IL-1β into its active form, IL-1β. Without caspase-1, Casp1⁻/⁻ mice are unable to generate active IL-1β via the canonical caspase-1-dependent pathway, though other pathways could contribute to activated IL-1β production. Casp1⁻/⁻ and wildtype mice were vaccinated with OmpX + LTA1 and subsequently challenged as described using a group of wildtype mice administered PBS as a control. Similar to observations in our Il1r1⁻/⁻ mice, we found that while vaccination increased CD4+ T cells and B cells in both Casp1⁻/⁻ mice and wildtype mice compared to the PBS-treated control, there was no difference in the number of either cell population between the vaccinated groups (Fig. 3a, b). These data suggest that the vaccine generated an adaptive immune response in both groups of vaccinated mice independent of caspase-1 activity on pro-IL-1β. When we compared the number of lung Th17 cells in the vaccinated wildtype and Casp1⁻/⁻ groups, we found that the absence of caspase-1 had no impact on the number of Th17 cells (Fig. 3c, d). Indeed, both immunized Casp1⁻/⁻ and wildtype mice had comparable numbers of Th17 cells in the lungs. To determine whether absence of caspase-1 affected vaccine efficacy, we challenged mice with K. pneumoniae and evaluated bacterial burdens in the lungs and spleens 24 h post-infection. Evaluation of bacterial burdens in

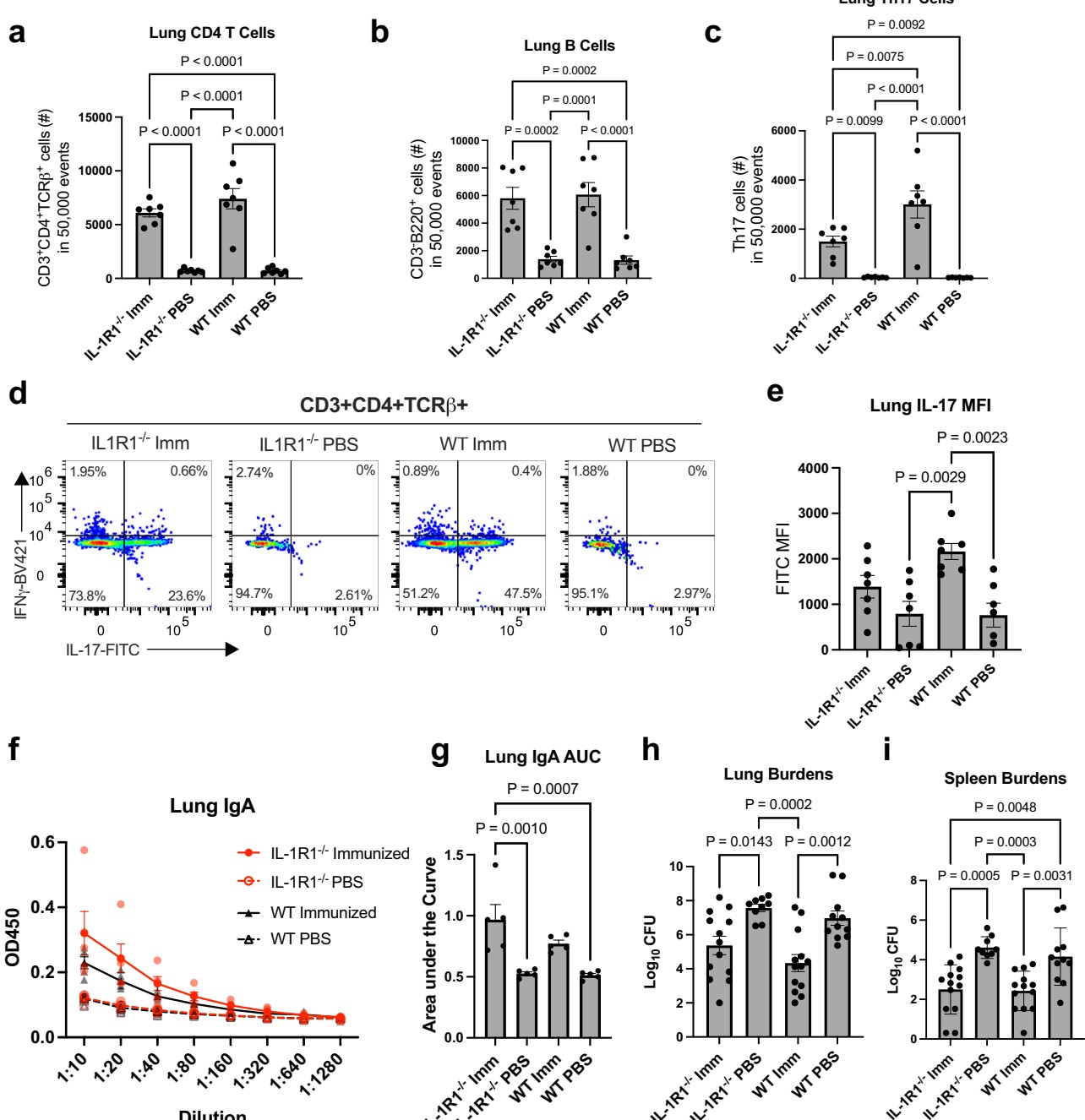

**Fig. 2 | IL-1 signaling promotes the generation of lung resident Th17 cells following mucosal immunization with OmpX + LTA1. a** CD4 T cell numbers, **b** B cell numbers, and **c** Th17 cell numbers out of 50,000 events from the lungs of vaccinated and unvaccinated wild-type and *Il1r1⁻/⁻* mice as measured in flow cytometry (*n* = 7 mice per group). **d** Representative dot plots for determining lung Th17 cells (CD3⁺, CD4⁺, TCR-β⁺, IL-17A⁺) in immunized and unimmunized wildtype and *Il1r1⁻/⁻* mice. **e** Median fluorescent intensity of FITC stained IL-17A in vaccinated and control mice (*n* = 7 mice per group). **f** IgA titers specific to heat killed

*K. pneumoniae* obtained from the supernatant of homogenized lungs in each group (*n* = 5 mice per group). **g** Area under the curve analysis of data displayed in *f*. Log transformed bacterial burdens in the (**h**) lungs and (**i**) spleen 24 h post-challenge in each group were measured in a cfu plating assay (*n* = 9 mice, IL-1R1⁻/⁻ PBS; *n* = 11 mice, WT PBS; *n* = 13 mice, IL-1R1⁻/⁻ Imm and WT Imm). All data are represented as mean +/− SEM. Statistical differences were determined using a one-way ANOVA followed by Tukey's multiple comparison test. Imm immunized.

the lungs revealed both immunized wildtype and *Casp1⁻/⁻* mice were protected from challenge and had on average 4-6 logs fewer CFU than the PBS control mice (Fig. 3e). In addition, there was no difference in bacteria burdens between the two vaccinated groups. Evaluation of spleen burdens determined that both vaccinated groups trended toward having fewer CFUs in the spleen compared to PBS control mice, indicative of less dissemination, though differences were not statistically significance (Fig. 3f). These data

demonstrate that caspase-1 dependent IL-1β is not required for the generation of lung Th17 cells or vaccine-mediated protection in our model.

**IL-1α enhances Th17 cell generation and promotes protection following vaccination with OmpX + LTA1**

While IL-1R1 signaling enhanced production of vaccine-elicited Th17 cells in the lungs, we were surprised IL-1β was dispensable in this process. Since

**Fig. 3 | Cleaved IL-1β is dispensable for vaccine mediated protection and expansion of Th17 cells.** **a** CD4 T cell numbers, **b** B cell numbers, and **c** Th17 cell numbers out of 50,000 events from the lungs of unvaccinated WT or vaccinated WT and *Casp1⁻/⁻* mice, as measured in flow cytometry. **d** Representative dot plots of Th17 cells (CD3⁺, CD4⁺, TCR-β⁺, IL-17A⁺) in each group. Log transformed bacterial burdens in the (**e**) lungs and (**f**) spleens 24 h post-challenge as determined using cfu plating assays. Data are presented as mean +/− SEM (*n* = 10 mice, *Casp1⁻/⁻* Imm; *n* = 8 mice, WT Imm; *n* = 6 mice, WT PBS). Statistical differences were determined using a one-way ANOVA followed by Tukey's multiple comparison test. Imm immunized.

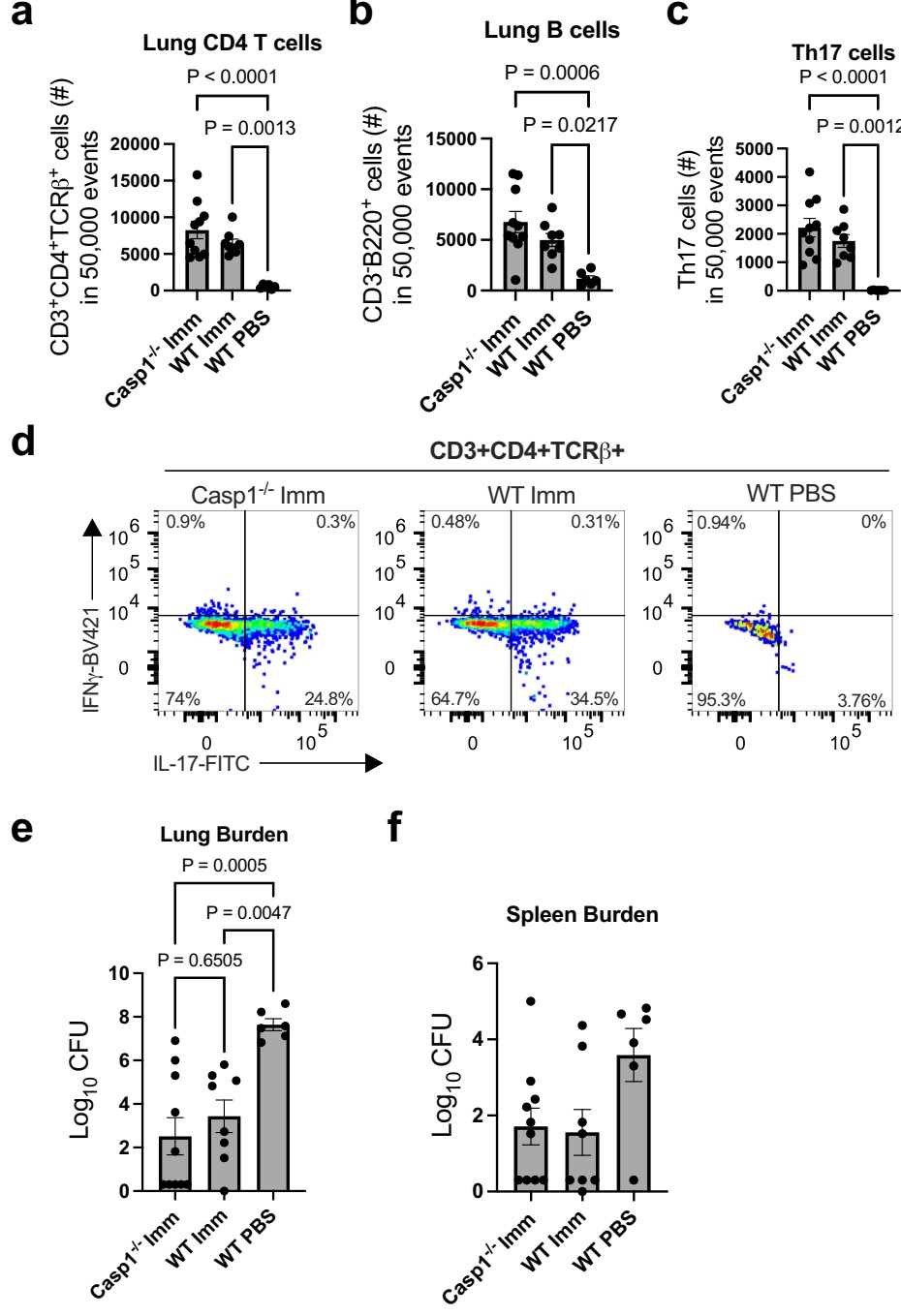

both IL-1α and IL-1β can signal through IL-1R1, we next investigated whether active IL-1α could be driving Th17 cell enrichment. We investigated the role of IL-1α using caspase-1 and caspase-4 double knockout mice (*Casp1/4⁻/⁻*). These mice are deficient in IL-1β and IL-18 and have a significant reduction in secreted IL-1α, though they are not IL-1α deficient[16]. To determine whether active IL-1α was necessary for generation of vaccine-mediated Th17 cells, *Casp1/4⁻/⁻* and wildtype mice were immunized with OmpX+LTA1 using PBS as a control. Vaccination resulted in enrichment of lung CD4+ T cells, B cells, and Th17 cells in both *Casp1/4⁻/⁻* and wildtype mice compared to their respective PBS treated controls (Supplementary Fig. 3a–d). To our surprise, there was no difference in these cell populations between immunized *Casp1/4⁻/⁻* and wildtype mice. Challenge of immunized mice with *K. pneumoniae* resulted in near identical lung burdens between the vaccinated wildtype and *Casp1/4⁻/⁻* mice 24 h post-infection

(Supplementary Fig. 3e). Interestingly, immunized *Casp1/4⁻/⁻* mice fared better than immunized wildtype mice against bacterial dissemination as demonstrated by spleen burdens 24 h post-infection (Supplementary Fig. 3f). Though both groups had significantly lower burdens than their respective PBS controls, we were unable to recover any bacteria from the spleens of immunized *Casp1/4⁻/⁻* mice, while immunized wildtype mice had an average of ~10² CFUs in the spleen. While reduction of IL-1α had no effect on Th17 cell enrichment in this model, it is possible levels of active IL-1α may still be sufficient in *Casp1/4⁻/⁻* mice for promoting a Th17 response upon vaccination.

We next used another mouse model deficient in IL-1α–*Il1a^{Δ559,1}* mice–to determine the role of this cytokine in eliciting Th17 cells upon vaccination. These mice were generated using CRISPR/Cas9 endonuclease-mediated gene editing and carry a 559 nucleotide deletion in Exon 3 of the

*Il1a* gene. As such, *Il1a*$^{\Delta559,1}$ mice have decreased secretory IL-1α levels compared to wildtype mice following 24 h or 48 h stimulation with LPS[17]. Wildtype and *Il1a*$^{\Delta559,1}$ mice were immunized and challenged as above and the immune response and protective efficacy of the vaccine were assessed. Similar to our observations in *Casp1*$^{-/-}$ and *Casp1/4*$^{-/-}$ mice, we found that immunized *Il1a*$^{\Delta559,1}$ mice had similar numbers of CD4+ T cells, B cells, and Th17 cells in the lungs compared to immunized wildtype mice (Supplementary Fig. 4a–c). Both vaccinated *Il1a*$^{\Delta559,1}$ and wildtype mice had significantly higher numbers of these lung cell populations than their PBS administered counterparts. Similar to the cellular response, the humoral response of immunized *Il1a*$^{\Delta559,1}$ and wildtype mice did not significantly differ upon vaccination. Both groups of immunized mice had elevated levels of OmpX specific serum IgG and lung IgA compared to their respective PBS administered controls (Supplementary Fig. 4d, e). Though lung IgA levels of immunized *Il1a*$^{\Delta559,1}$ mice trended to being higher than those of immunized wildtype mice, the differences were not statistically significant (Supplementary Fig. 4e). When examining bacterial burdens in the lung and spleen following pulmonary challenge with *K. pneumoniae*, we found that both immunized *Il1a*$^{\Delta559,1}$ and wildtype mice had reduced burdens in the lungs and spleens compared to PBS treated mice (Supplementary Fig. 4f, g).

Given *Il1a*$^{\Delta559,1}$ may not result in complete absence of active IL-1α we tested serum levels of IL-1α before and 24 h after pulmonary challenge with *K. pneumoniae* in knockout and wildtype mice. Additionally, 24 h post-challenge, we collected and homogenized the lungs of *Il1a*$^{\Delta559,1}$ and wildtype mice and tested the supernatant for IL-1α using ELISA. As expected, the *Il1a*$^{\Delta559,1}$ mice had no detectable IL-1α in the serum prior to infection, and a small, though not statistically significant, increase 24 h after infection. Conversely, the wildtype mice had a dramatic rise in serum IL-1α following infection (Supplementary Fig. 4h). When evaluating IL-1α in lung homogenate supernatant following infection, we found that both *Il1a*$^{\Delta559,1}$ and wildtype mice had elevated levels of IL-1α that were not significantly different from one another (Supplementary Fig. 4i). Given the ELISA may not distinguish between IL-1α and its precursor pro-IL-1α, it is possible that the elevated levels measured were a result of release of the pro-IL-1α from homogenized tissue. However, *K. pneumoniae*-induced damage to lung tissue could also result in release of pro-IL-1α, which has full bioactivity and can trigger IL-1R1 signaling-dependent inflammation during infection[18]. These data suggest that while the *Il1a*$^{\Delta559,1}$ mice have significantly impaired systemic IL-1α production, this impairment may not be sufficient for reducing activity of pro-IL-1α released from damaged tissues. This possibility offers an explanation as to why we were unable to replicate the phenotype of reduced numbers of Th17 cells in the lungs of immunized *IL1R1*$^{-/-}$ with the *Il1a*$^{\Delta559,1}$ mice.

While we were unable to find a true IL-1α deficient genetic model, we next used antibody neutralization to investigate the role of IL-1α in enhancing the tissue resident Th17 population. To neutralize IL-1α, wildtype mice immunized as described were injected interperitoneally with anti-IL-1α, anti-IL-1β, or isotype control antibodies one day prior to receiving their 2$^{nd}$ vaccine dose. Following challenge as above, we observed that neutralization of IL-1 signaling had no impact on the numbers of CD4+ T cells and B cells in the lungs following vaccination (Fig. 4a, b). However, treatment with anti-IL-1α or anti-IL-1β resulted in a reduction in the number of lung Th17 cells compared to isotype control-treated mice (Fig. 4c, d). These data support our observations in Fig. 2 demonstrating a role for IL-1R1 signaling in vaccine-mediated Th17 cell production. Analysis of whole lung single cell suspensions using IL-17A ELISpot revealed there was no difference in the number of IL-17A secreting cells from vaccinated mice regardless of antibody treatment (Fig. 4e). Given these data represent the whole lung, including γδ T cells and/or type 3 innate lymphoid cells (ILC3s) which also secrete IL-17A, differences in IL-17A production from Th17 cells could be obscured. In addition, mice treated with IL-1α or IL-1β neutralizing antibodies had levels of OmpX specific serum IgG comparable to vaccinated mice treated with isotype control antibody, suggesting inhibition of IL-1 signaling had no impact on vaccine elicited antibody responses (Fig. 4f, g).

We next evaluated how neutralization of IL-1 signaling influenced protective efficacy of our vaccine. Upon examining lung burdens of challenged mice, we found that treatment with anti-IL-1α resulted in significantly higher bacterial burdens than the isotype-treated vaccinated mice and comparable burdens to unvaccinated naïve mice (Fig. 4h). Interestingly, vaccinated mice treated with IL-1β neutralizing antibody retained a level of protective efficacy with significantly lower lung burdens than the naïve mice and no significant difference from isotype-treated vaccinated mice. The evaluation of spleen bacterial burdens demonstrated a more dramatic effect of IL-1 neutralization on bacterial dissemination. Similar to the lungs, neutralization of IL-1α signaling eliminated the protective efficacy of our vaccine and there was no significant difference between naïve mice and anti-IL-1α-treated vaccinated mice (Fig. 4i). Unlike our observations with lung burdens, vaccinated mice treated with IL-1β neutralizing antibody had significantly higher spleen burdens than the vaccinated isotype treated controls.

### IL-1α as a vaccine adjuvant generates a protective immune response

Since our data demonstrate that IL-1-mediated signaling influences generation of Th17 cells and protection in our vaccine model, we next examined whether IL-1α or IL-1β could be used as a vaccine adjuvant to confer immunity and protection in our model. To test this, we immunized mice following the previously described schedule replacing the LTA1 adjuvant with either recombinant IL-1α or IL-1β. Upon challenging vaccinated mice with *K. pneumoniae*, we found that mice adjuvanted with either rIL-1α or rIL-1β had significantly reduced lung bacterial burdens compared to PBS administered controls, with the rIL-1α and rIL-1β groups reducing the burdens by 4 and 3 logs respectively (Fig. 5a). Protection from dissemination, as indicated by bacterial burdens in the spleen, were not as clear. There was no difference between IL-1β adjuvanted mice and PBS controls. Additionally, 3 out of 7 of the IL-1α adjuvanted mice were protected from bacterial dissemination as evidenced by having no detectable bacteria in the spleen, while the remaining 4 mice had similar burdens to PBS treated mice. However, differences between PBS administered and rIL-1α adjuvanted groups remained significant (*P* = 0.0262, Fig. 5b). Further, we found that rIL-1α and rIL-1β as adjuvants were sufficient in generating adaptive immune responses. Both vaccination groups generated high levels of OmpX-specific serum IgG and elevated lung IL-17A-secreting T cells when compared with the PBS administered group, as evidenced by ELISA and ELISpot, respectively (Fig. 5c, d). To confirm the increase of Th17 cells observed in IL-1α + OmpX vaccinated mice was not due to infection, we also performed ELISpot on lung cells isolated from uninfected immunized or PBS-treated mice. Vaccination alone resulted in an increase in Th17 cells, and abrogation of IL-1 signaling in *Il1r1*$^{-/-}$ mice resulted in a dramatic decrease in the observed number of Th17 cells (Supplementary Fig. 5).

### Discussion

We have demonstrated that signaling through IL-1R1 plays a contributing role in vaccine-mediated Th17 cell generation, the majority of which we have found to be lung TRM cells[3]. Analysis of sc-RNA-seq data revealed that these cells express high levels of IL-1R1 compared to naïve CD4+ T cells. Cytokine analysis in the BAL of immunized mice showed vaccination induced both IL-1α and IL-1β. Somewhat surprisingly, vaccine driven lung Th17 cells were generated independent of caspase-1, demonstrating that inflammasome cleavage of IL-1β or IL-18 is dispensable. However, neutralization of IL-1α was associated with reduced lung Th17 TRM generation. Moreover, recombinant IL-1α was somewhat more effective than IL-1β as a cytokine adjuvant to generate lung Th17 cells. Taken together, these data suggest that IL-1α is a major downstream event for LTA1 to induce lung Th17 TRM cells and suggest that IL-1α may serve as a cytokine adjuvant to generate these types of immune responses in the lung.

Elucidation of mechanisms underlying Th17 TRM development from naïve T cells is critical for design and implementation of new mucosal adjuvants. The results reported herein demonstrate that an LTA1

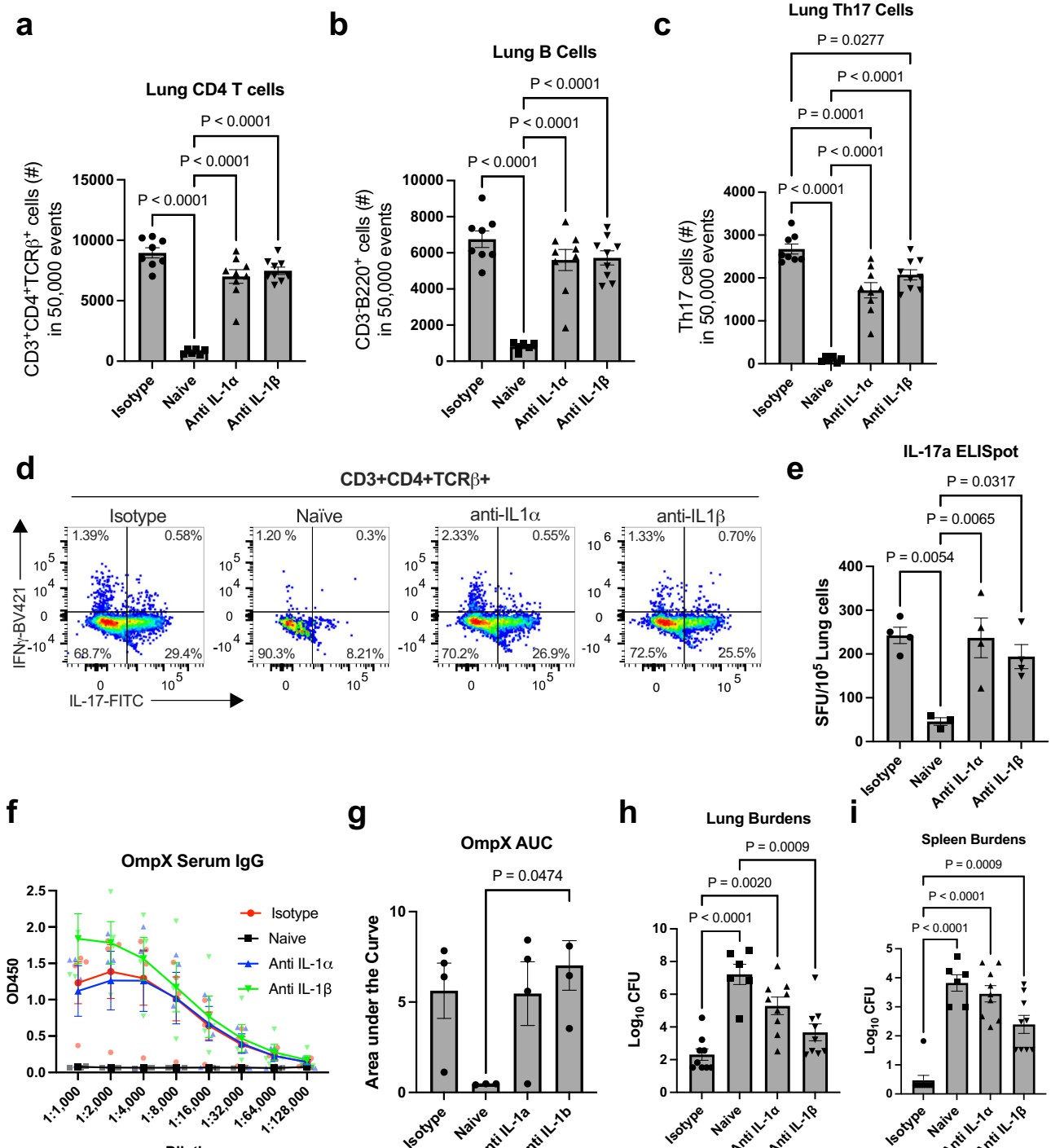

**Fig. 4 | Antibody neutralization of IL-1α and to a lesser degree IL-1β prevents vaccine-mediated protection and Th17 cell generation.** **a** CD4 T cells, **b** B cells, and **c** Th17 cells in 50,000 events from the lungs of vaccinated and unvaccinated mice either untreated (Naïve) or treated with isotype antibody, anti-IL-1α, or anti-IL-1β, as measured in flow cytometry ($n = 8$ mice, Isotype; $n = 6$ mice, naïve; $n = 9$ mice, anti-IL-1α and anti-IL-1β). **d** Representative dot plots of Th17 cells (CD3[+], CD4[+], TCR-β[+], IL-17A[+]) in untreated (Naïve) and antibody- or isotype control-treated mice. **e** ELISpot measuring IL-17A secreting cells after overnight stimulation with OmpX. Data are displayed as spot counts (SFU) per $10^5$ plated cells from single cell lung suspensions ($n = 3$ mice, naïve; $n = 4$ mice for all other groups). **f** Serum IgG titers specific to OmpX from each vaccinated group ($n = 3$ mice, naïve; $n = 4$ mice for all other groups). **g** Area under the curve of the data depicted in *f*. Log transformed bacterial burdens in the (**h**) lungs and (**i**) spleens 24 h post-challenge were determined using cfu plating assays ($n = 6$ mice, naïve; $n = 9$ mice, all other groups). All data are presented as mean +/− SEM. Statistical differences were determined using a one-way ANOVA followed by Tukey's multiple comparison test.

adjuvanted vaccine activates IL-1 signaling pathways to elicit Th17 TRM cells. These data are consistent with previous reports suggesting parenterally administered LT also elicits Th17 cells via signaling through IL-1R1[19]. While previous studies have shown LTA1 activates NLRP3, cleavage of pro-caspase-1 into caspase-1, and secretion of IL-1β in vitro[4], IL-1β appeared to be dispensable for Th17 cell generation in our model. Instead, we found that our LTA1 adjuvanted vaccine acted through IL-1α signaling, in part, to promote antigen-specific Th17 cells, in line with previously demonstrated functions of IL-1α in promoting activation and expansion of memory T cells[10,20]. Indeed, antibody neutralization of IL-1α resulted in a decrease of

**Fig. 5 | IL-1α is sufficient as a vaccine adjuvant to generate protective immunity against *K. pneumoniae.*** Log transformed bacterial burdens in the (**a**) lungs and (**b**) spleen 24 h post-challenge of C57Bl/6 mice immunized as indicated ($n = 7$, OmpX +rIL-1α; $n = 3$, OmpX+rIL-1β; $n = 8$, PBS). **c** Serum IgG titers specific to OmpX from each vaccinated group as measured in ELISA ($n = 3$ mice per group). **d** ELISpot measuring IL-17A secreting cells after overnight stimulation with OmpX. Data are displayed as spot counts (SFU) per $10^5$ plated cells from lung homogenate ($n = 2$ mice per group). All data are presented as mean +/− SEM. Statistical differences were determined using a one-way ANOVA followed by Tukey's multiple comparison test.

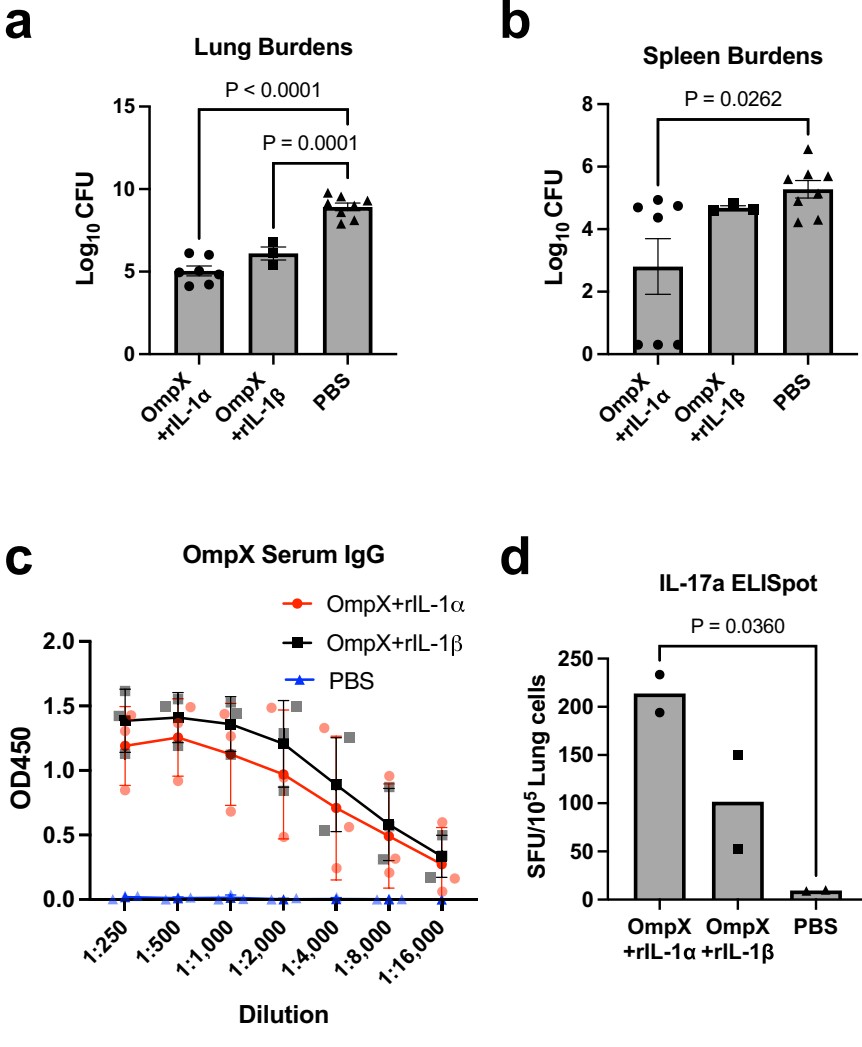

vaccine-induced Th17 cell generation coupled with a reduction in vaccine-mediated protection (Fig. 4). However, animal models knocking down IL-1α expression in vivo failed to demonstrate a phenotype in our studies. We expect this is due to incomplete suppression of IL-1α release in *Casp1/4*[−/−16] and *Il1a*[Δ559,1] mice, where we observed similar levels of *K. pneumoniae*-induced IL-1α in the lungs of wildtype and *Il1a*[Δ559,1] mice (Supplementary Fig. 4). These results suggest even low levels of IL-1α could promote Th17 cell formation. Importantly, these studies could benefit from a more recently developed IL-1α -deficient mouse model, which does not affect IL-1β and could potentially better illustrate the role of IL-1α in vaccine-mediated Th17 cell generation[21]. Given we did not observe a complete abrogation of Th17 production upon neutralization of IL-1α, it is possible this cytokine is not the only factor facilitating development of Th17 TRMs from naïve T cells. Thus, the roles additional cytokines or other signaling pathways play in vaccine-mediated elicitation of memory T cells remain to be determined.

That IL-1β appears to be dispensable for Th17 cell development following immunization with OmpX + LTA1 (Fig. 3) was unexpected given the well-established role for IL-1β in Th17 cell development in mice and humans[5,7–11,22,23]. The LTA1-related adjuvants LT and double-mutant LT (dmLT) have also been shown to stimulate Th17 cells through IL-1β and IL-23 related mechanisms upon stimulation of human cells in vitro or following parenteral immunization in mice[19,24,25]. It is difficult to rule out a role for IL-1β signaling in Th17 cell generation following vaccination with OmpX + LTA1 given our mouse models only looked at inhibition of the canonical caspase-1-dependent pathway for IL-1β cleavage. Thus, to more precisely

determine the role of active IL-1β in LTA1-mediated Th17 generation, future studies should examine more complete inhibition of all IL-1β cleavage pathways. Indeed, somewhat contradictory to our animal data in Fig. 3, antibody neutralization of IL-1β resulted in a reduction in lung Th17 cell generation following vaccination, though these mice still had reduced burdens in the lung following *K. pneumoniae* challenge (Fig. 4). One possible explanation for these discrepancies is that IL-1α, but not IL-1β, is necessary to promote protective effector cytokine production (i.e., IL-17A, IL-17F, and IL-22) from Th17 cells elicited by mucosal administration of an LTA1 adjuvanted vaccine[11].

Importantly, our results suggest IL-1α could be an effective mucosal adjuvant for eliciting both cellular and humoral responses to vaccine antigens in vivo (Fig. 5). Identification of such adjuvants is critical to development of more broadly effective vaccines. Our data are in line with previous reports demonstrating the adjuvanticity of IL-1 cytokines. Both IL-1α and IL-1β have been shown to elicit antigen-specific antibodies when administered with protein antigens via either mucosal or parenteral routes[26–28], and IL-1α administered subcutaneously was shown to mildly enhance T helper cell generation in vivo[29,30]. In addition, an influenza vaccine adjuvanted with either IL-1α or IL-1β resulted in antigen-specific protection in a mouse model[31], while another study found IL-1β adjuvanted influenza vaccine induced protective TRMs[32]. Our study expands these findings to demonstrate IL-1α as a mucosal adjuvant can elicit both antigen-specific antibodies and lung Th17 TRMs when admixed with OmpX, providing protection against subsequent *K. pneumoniae* challenge.

One limitation of our study is that the knockout mouse models used in this study likely have global effects on immune signaling outside of vaccine-elicited changes. We attempted to address this limitation by using transient inhibition of IL-1 signaling (Fig. 4) as well as demonstrating the ability of IL-1 adjuvants to elicit Th17 cell generation (Fig. 5). However, future studies should further tease apart the role of IL-1 signaling by making use of inducible cell-specific knockout models of the inflammasome or IL-1 signaling as well as gain-of-function studies. Another limitation of this study is that while IL-1R1 appears to be important for eliciting vaccine-induced protective Th17 cells, it does not appear to be the only factor leading to Th17 cell production. Elucidating additional pathways leading to vaccine-mediated Th17 cells will further improve mucosal vaccine development. For example, additional inflammasome-mediated cytokines, such as the IL-1 related cytokine IL-18, could be evaluated. While we found that mice deficient in caspase-1, which cleaves pro-IL-18 into its active form, did not show a deficiency in production of vaccine-generated Th17 cells, non-canonical pathways for IL-18 activation could be impacting these results. Importantly, Th17 cells produced in vaccinated *Il1r1*$^{-/-}$ mice still provided protection upon subsequent challenge, so a future focus will be to determine whether T cells generated in wild-type and *Il1r1*$^{-/-}$ mice are functionally different. In addition, we do not know the longevity of Th17 cells elicited by an IL-1α adjuvant. Thus, future studies will examine the lifetime of these cells in immunized mice to determine whether they can provide long-lasting protection.

Overall, we have demonstrated IL-1α signaling is important for vaccine-mediated generation of Th17 TRMs in mice. In addition, IL-1α and, to a lesser extent, IL-1β appear to be effective on their own to generate these cells when admixed with our *K. pneumoniae* antigen OmpX. Our study sheds light on mechanisms by which LTA1 induces vaccine-mediated protection and further highlights the potential of IL-1 cytokines as mucosal vaccine antigens. Overall, an increased understanding of the adjuvant-dependent mechanisms through which naïve T cells develop into Th17 TRM cells will ultimately enable development of improved mucosal vaccines.

## Methods
### Mice
For all in vivo studies, male mice aged 7–10 weeks on a C57Bl/6J background were used. Wildtype C57Bl/6 mice used in the study were either bred in house or purchased from the Jackson Laboratory. Transgenic mouse strains *Il1r1*$^{-/-}$ (Strain #:003245)[14], *Casp1*$^{-/-}$ (Strain #:032662)[15], *Casp1/4*$^{-/-}$ (Strain #:016621)[16], and CB57BL/6J-*Il1a*$^{em2Lutzy}$/Mmjax (*Il1a*$^{A559,1}$, Strain #: 067031-JAX)[17] were purchase from the Jackson Laboratory. CB57BL/6J-*Il1a*$^{em2Lutzy}$/Mmjax was obtained from the Mutant Mouse Resource and Research Center (MMRRC) at The Jackson Laboratory, an NIH-funded strain repository, and was donated to the MMRRC by Cat Lutz, Ph.D., The Jackson Laboratory. All mice were housed in specific pathogen free conditions at Tulane University School of Medicine and provided food and water *ad libitum*. Experimental procedures were conducted in accordance with protocols approved by Tulane University's Institutional Animal Care and Use Committee. We have complied with all relevant ethical regulations for animal use.

### Antigen and adjuvant preparation and vaccination strategy
The vaccine antigen, recombinant OmpX, was produced and purified using an *Escherichia coli* expression system as previously described[3]. The adjuvant LTA1 was isolated and purified using previously described methodologies[2,33]. Recombinant IL-1α and IL-1β used as adjuvants were purchased from R&D Systems (Cat #400-ML-005/CF and 401-ML-010/CF). Mice were vaccinated following the timeline and dosing as described previously[3]. In brief, mice were vaccinated following a prime/boost strategy, with the boost occurring 21 days following the prime. Each vaccination dose consisted of 1 µg OmpX mixed with 10 µg LTA1 in a 50 µL volume of PBS. For our studies using rIL-1α and rIL-1β as adjuvants, 1 µg of OmpX was mixed with 2.5 µg of each respective cytokine in a 50 µL volume of PBS. Each vaccine was administered by direct instillation into the lungs via oropharyngeal aspiration following anesthetization with isoflurane[34]. For all challenge studies, mice were challenged 7–10 days following the vaccine boost.

### Oropharyngeal infection with *K. pneumoniae* and enumeration of bacterial burdens
*K. pneumoniae*-396 (K1) was grown overnight in a 3 mL culture of luria broth (LB) Miller (VWR) or tryptic soy broth (Difco) at 37 °C with shaking at 233 rpm. Overnight cultures were then subcultured with a 1:100 dilution into 30 mL of the same media. This subculture was grown in the same conditions for 2.5 h to achieve early logarithmic growth phase. The concentration of *K. pneumoniae* was determined by reading the optical density at 600 nm. The bacteria were then pelleted via centrifugation at ~5000 × *g* for 8 min and washed 2× with sterile PBS. Bacteria were resuspended in sterile PBS to the desired concentration. To confirm the dose of bacteria, the inoculum was serially diluted in PBS and spot plated on LB agar plates to count CFUs. For infection, mice were anesthetized by isoflurane inhalation and 1 ×10$^4$ CFU of K1 was directly instilled into the lungs via oropharyngeal aspiration in a 50 µL volume. To enumerate bacterial burdens, mice were euthanized via CO$_2$ asphyxiation at 24 h post infection. The right lung and spleen were collected and placed in their own tubes of 1 mL of sterile PBS on ice until further processing. The organs were subsequently homogenized using a Multi-Gen 7XL handheld tissue homogenizer (Proscientific). Homogenized tissue was serially diluted in sterile PBS and spot plated on LB agar plates for CFU counts.

### Generation of single cell suspension for flow cytometry and ELISpot
The left lung of euthanized mice was collected in 700 µL sterile PBS and kept on ice for further processing. The PBS was decanted, and the tissue was minced manually with dissection scissors. Minced tissue was resuspending in 2 mL IMDM (Gibco) containing 2 mg/mL collagenase (Sigma-Aldrich) and 80 U/mL DNase1 (Sigma-Aldrich) and incubated at 37 °C with shaking at 233 rpm for 1 h. Digested tissue was passed through a 70 µm cell strainer (Fisher) and red blood cells were removed using ACK lysis buffer (Gibco). Isolated cells were resuspended in 1 mL IMDM containing 10% FBS (Hyclone) and counted on a Cellometer for downstream applications.

### Flow cytometry and intracellular cytokine staining
For IL-1R1 expression, single cell suspensions from PBS treated or immunized mice and prepared as above were washed with FACS buffer (1× PBS with 0.5% BSA) followed by Fc receptor blocking (BD, Cat #553142) for 1 h at 4 °C. Cells were then stained with PE hamster anti-mouse cd121a (clone JAMA-147, BioLegend, Cat# 113505) for 20 min at 37 °C prior to adding additional cell markers including AlexaFluor 647 mouse anti-mouse CD45.2 (clone 104, BioLegend, Cat #109818), PE-Cy5 hamster anti-mouse CD3e (clone 145-2C11, BD, Cat #561825), BV480 rat anti-mouse CD19 (clone 1D3, Invitrogen, Cat #414-0193-82), and PerCP-Cy5.5 rat anti-mouse CD4 (clone RM4-5, BioLegend, Cat #100540) and further incubating for 1 h at 4 °C. After incubation, cells were fixed and permeabilized with cytofix/cytoperm (BD Biosciences) for 20 min, washed 3x, resuspended in FACS buffer, and analyzed using a Cytek Aurora spectral flow cytometer.

For intracellular cytokine staining, single cell suspensions were made of lungs collected from vaccinated and infected mice and 1 ×10$^6$ cells from each sample were added to the wells of a 96-well round bottom plate. Cells were stimulated with 50 ng/mL of Phorbol 12-myristate 13-acetate (Sigma-Aldrich) and 750 ng/mL ionomycin (Sigma-Aldrich) for 5 h at 37 °C. At 1 h of stimulation, 1 mg/mL GolgiStop (BD Bioscience) was added to prevent the secretion of cytokines. Cells were washed with FACS buffer (1× PBS with 0.5% BSA) and fixed and permeabilized with cytofix/cytoperm (BD Biosciences) for 20 min. Cells were washed with 1× Perm/wash buffer (BD Biosciences), Fc receptor blocked, and stained for 30 min for desired cell markers. Cells were washed 3x, resuspended in FACS buffer, and analyzed using a Cytek Aurora spectral flow cytometer. For each sample, 5 ×10$^4$ events were recorded and all analysis was conducted using FlowJo software

version 9 (Tree Star). Antibodies used for blocking and staining are as follows: Rat Anti-Mouse CD16/CD32 Fc Block (clone 2.4G2, BD Biosciences, Cat #553141), PE-Cy7 Rat Anti-Mouse CD4 (clone RM4-5, BD Biosciences, Cat #561099), APC rat anti-mouse CD3e (clone 17A2, BioLegend, Cat #100236), PE-Cy5 hamster anti-mouse TCRβ (clone H57-597, BD Biosciences, Cat #553173), FITC rat anti-mouse IL-17A (clone TC11-18H10.1, BioLegend, Cat #506907), Brilliant Violet 421 rat anti-mouse IFNγ (clone XMG1.2, BioLegend, Cat #505829), PE rat anti-mouse B220 (clone RA3-6B2, BioLegend, Cat #103207).

## Antibody neutralization of IL-1α and IL-1β

Wild type male C57Bl/6J mice aged 8 weeks were vaccinated with OmpX + LTA1 as described above. At 1 day prior to boost, mice were with treated with 200 μg of anti-IL-1α, anti-IL-1β, or anti-IL-1α+ anti-IL-1β neutralizing antibodies or isotype control via intraperitoneal injection in a 200 μL volume. One group of naïve mice were used as an additional control. Vaccinated and antibody treated mice were challenged and euthanized following the vaccination timeline described above. All antibodies were diluted in sterile PBS prior to injection. Neutralizing antibodies were purchased from Bio X cell: anti-IL-1α (clone ALF-161, Cat #BE0243), anti-IL-1β (clone B122, Cat #BE0246), and isotype control (polyclonal Armenian hamster IgG, Cat #BE0091).

## In vivo inflammatory cytokine analysis

Mice were vaccinated following the prime/boost schedule described above with one group receiving OmpX + LTA1 and the other sterile PBS. At 24 h following the boost, all mice were euthanized and serum, bronchial alveolar lavage fluid (BALF), and supernatant from whole lung homogenate were collected for cytokine analysis. In brief, blood was collected via cardiac puncture and spun at 2500 rpm on a benchtop microcentrifuge for 20 min to collect serum. For BALF, cOmplete ULTRA tablets, mini, EASYpack Protease Inhibitor cocktail (Roche, Cat #5892970001) was prepared in PBS following manufacturer's instructions. This solution was instilled into the lungs of mice intratracheally at a 1 mL volume for 3 washes and kept on ice. Whole lungs were then collected in 1 mL of the protease cocktail, homogenized with the handheld tissue homogenizer as described, pelleted at 2500 rpm for 15 min at 4 °C, and the supernatant was collected. Inflammatory cytokines were quantified from each sample using a LegendPlex mouse inflammation panel (BioLegend, Cat #740150) following manufacturer's instructions. Samples were processed using a Cytek Aurora spectral flow cytometer and analyzed using the LEGENDplex™ Data Analysis Software Suite (Qognit).

## ELISA

ELISAs were performed to evaluate serum IgG and lung IgA titers following previously published methods[3]. Serum was collected as described above. Lung IgA was collected in BALF consisting of 1 mL sterile PBS and 3 washes of the lungs. 96 well plates were coated overnight with 0.1 μg heat killed K2 strain *K. pneumoniae* or rOmpX in 100 μL per well. Coated plates were washed with washing buffer (0.05% Tween 20 in PBS) and blocked for 2 h with blocking buffer (1% bovine serum albumin and 0.1% Tween 20 in PBS). Following blocking, plates were washed 3× and serially diluted serum or BALF was added. Plates were left to incubate at room temperature for 2 h and washed 5x. Bacterial specific antibodies were detected using 1:4000 diluted goat anti-mouse IgG or IgA conjugated with horseradish peroxidase (Southern Biotech, Cat #1036-05 and 2050-05) and incubated for 1 h at room temperature. Plates were washed 5× and 3,3',5,5'-tetramethyl-benzidine peroxidase substrate (TMB, ThermoFisher, Cat #N301) was added to each well. Absorbance was read at 450 nm on a 96 well plate reader (Biotek). For the quantification of IL-1α, we used the Mouse IL-1alpha ELISA MAX Deluxe kit from BioLegend following the manufacturer's recommendations (Cat #433404)

## ELISpot

ELISpot was used in some experiments for the quantification of IL-17A producing cells in the lungs of immunized mice. To do this Millipore

MultiScreen-IP plates (Millipore Sigma, Cat #MAIPS4510) were activated with 50 μL/well freshly prepared 70% ethanol for two minutes. Plates were washed 4× with PBS and coated with 2.5 μg/mL anti mouse IL-17A antibody (clone 50101, R&D, Cat #MAB721-100) in PBS at 4 °C overnight. Plates were washed 4× with wash buffer (1× PBS with 0.05% Tween 20) and incubated with complete IMDM (IMDM with 10% BSA) for 2 h at 37 °C. Following incubation, IMDM was removed and $1 \times 10^5$ isolated lung cells and 2 μg/mL OmpX were added in triplicate to a total volume of 100 μL per well. Cells were then incubated at 37 °C with 5% $CO_2$ for 18 h. Following incubation, plates were washed 4× with wash buffer and biotinylated anti-mouse IL-17A antibody (polyclonal, R&D, Cat #BAF421) was added in assay buffer (1× PBS, 0.05% Tween 20, 0.5% BSA) at a concentration of 0.8 μg/mL in a volume of 100 μL per well. Plates were incubated for 2 h with gentle shaking at room temperature. Plates were washed 4× and incubated with 1:2000 diluted streptavidin-alkaline phosphatase (R&D, Cat #AR001) for 45 min with gentle shaking at room temperature. Plates were washed once more and spots were developed using BCIP/NBT substrate solution (Sigma-Aldrich, Cat #B5655) for 15 min. After development, plates were read on a CTL ImmunoSpot S4 and analyzed with ImmunoSpot Software (Cellular Technology Ltd) for the quantification of spot forming units (SFUs).

## Single-cell RNA sequencing

Single-cell RNA sequencing was published previously[3]. Essentially, $1 \times 10^6$ cells were collected as whole-lung or whole-spleen single-cell populations. Cells were subjected to enrichment by using a CD4 positive selection kit (catalog no. 130-104-454, Miltenyi Biotec) and treated with 100 μl of TrypLE for 1 min to dissociate single cells from small aggregates or clusters. Cell numbers and viability were validated by Cellometer Auto 2000 (Nexcelom Bioscience) before preparation of scRNA-seq library. For 10× single-cell 3′ RNA-seq assay, 5000 live cells per sample were targeted by using 10× scRNA-seq technology provided by 10X Genomics Inc. (CA, USA). Briefly, viable single-cell suspensions were partitioned into nanoliter-scale Gel Beads-In-EMulsion (GEMs). Full-length barcoded complementary DNAs (cDNAs) were then generated and amplified by PCR to obtain sufficient mass for library construction. After enzymatic fragmentation, end-repair, A-tailing, and adapter ligation, single-cell 3′ libraries comprising standard Illumina P5 and P7 paired-end constructs were generated. Library quality controls were performed by using an Agilent High Sensitive DNA kit with Agilent 2100 Bioanalyzer and quantified with a Qubit 2.0 fluorometer. Pooled libraries at a final concentration of 1.8 pM were sequenced with paired-end single index configuration by Illumina NextSeq 550. Cell Ranger version 2.1.1 (10X Genomics) was used to process raw sequencing data and Loupe Cell Browser (10X Genomics) to obtain differentially expressed genes between specified cell clusters. In addition, Seurat suite version 2.2.1[35] was used for quality control and downstream analysis. Filtering was performed to remove multiplets and broken cells. Also, uninteresting sources of variation were regressed out. Variable genes were determined by iterative selection based on the dispersion versus average expression of the gene. For clustering, principal components analysis was performed for dimension reduction. The top 10 principal components were selected by using a permutation-based test implemented in Seurat and passed to t-SNE for clustering visualization. Gene Expression Omnibus accession number is GSE178385.

## Statistics and reproducibility

All statistical analysis was conducted using GraphPad Prism (version 9). For analysis comparing two groups, a Student's T test was used. For comparisons of 3 or more groups, we used one-way analysis of variance (ANOVA) with Tukey's post hoc analysis after applying the Bonferroni correction for multiple comparisons. For all analyses involving bacterial burdens, we performed a log transformation on the data and performed ANOVA on transformed data as described above. Areas under the curve were determined using the 'Area under curve' analysis in GraphPad Prism. All

data points represent a measurement taken from distinct samples. Experiments were performed in duplicate and sample sizes are indicated in each figure. *P*-values for significant differences are provided on each graph and/or annotated as follows: *\*P* < 0.05, *\*\*P* < 0.01, *\*\*\*P* < 0.001, *\*\*\*\*P* < 0.0001.

## Study approval

Experimental procedures were conducted in accordance with protocols approved by Tulane University's Institutional Animal Care and Use Committee.

## Reporting summary

Further information on research design is available in the Nature Portfolio Reporting Summary linked to this article.

## Data availability

The sequencing data in Fig. 1 have been deposited in NCBI's Gene Expression Omnibus and can be accessed using accession number GSE178385. The source data behind the graphs in the paper can be found in Supplementary Data 1 or is available upon request.

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

## Acknowledgements

This work was supported by the Louisiana Board of Regents Support Fund LEQSF(2021-24)-RD-A-19 to J.E.M., and the Louisiana Board of Regents Endowed Chairs for Eminent Scholars program, as well as by PHS grant R35HL139930 and NIAID/NIH awards 5R01AI149119 to J.K.K. We would like to thank Alanna G. Wanek for help in processing of sc-RNA-seq data.

## Author contributions

J.P.H., J.K.K, and J.E.M. designed research studies; J.P.H., A.S., H.Y., N.I., T.P.R., J.L.H. and R.S. performed experiments and acquired data; K.S. performed RNA sequencing experiments; E.B.N. provided LTA1; and J.P.H., J.K.K. and J.E.M analyzed data and wrote the manuscript.

## Competing interests

The authors declare the following competing interests: Tulane University has filed a PCT application to the U.S. Patent and Trademark Office on the vaccine described here.
