## [Peer Review File · Communications Biology]

Reviewers' comments:

Reviewer #1 (Remarks to the Author):

The manuscript by Hoffman et al. examines the contribution of IL-1alpha and IL-1beta in the generation of protective immunity through the vaccine-elicited activation of lung tissue resident memory (Trm) cells producing IL-17A. Their work implicates that IL-1alpha is both necessary and sufficient to elicit robust vaccine responses in their model system.

The studies conducted are appropriate based on widely-available tools, conveyed clearly, and presented in a straightforward manner. There are a few comments and questions that the authors could address to improve the manuscript for readers.

1. In the Introduction, describe the immunization method/site used in prior and current studies. Later reading indicates the immunization is via oropharyngeal aspiration to the lung, but it would be valuable to know this earlier. Also, consider including this information in the first paragraph of the Results.
2. Lines 80-81 allude to low Il1r1 expression on naïve lung resident CD4+ T cells, but the data in Figure 1B are a comparison of lung (activated) and spleen (naïve) CD4+ T cells.
3. As another inflammasome-regulated cytokine, was IL-18 measured in the samples used for Figure 1C-N?
4. Line 17 contains an extra "role".
5. It seems unlikely the IL-1alpha ELISA would be able to discriminate processed from unprocessed IL-1a in the lung tissue, whereas it could be assumed that the IL-1a in the serum is the processed form. Perhaps the local IL-1a in the lung comes about as a consequence of processing-independent release from damaged epithelial cells (wherein it is described as a chromatin-associated nuclear protein). This could be discussed.
6. Are Trm cells increased in the IL-1a-adjuvanted immunization protocol and are these cells decreased in the mice in which IL-1a (or IL-1b) have been inhibited?
7. The authors could mention the potential of using the newly-described IL-1alpha-deficient mice generated by Dr. Kanneganti's group (PMC9729281) to provide additional insight into the contribution of IL-1a in this vaccine model.

Reviewer #2 (Remarks to the Author):

The authors, Hoffmann and colleagues, described that immunization of OmpX from *K. pneumoniae* with adjuvant LTA1 induces Th17 mediated by IL-1 signaling. They proposed the underlying mechanism by

which the mixture of adjuvant LTA1 and antigen OmpX elicit antigen specific lung Th17 cells and further suggested the role of IL-1a as an adjuvant. Overall, this is well studied about the importance of IL-1 in the Th17 induced by vaccination. However, there are several points which need to be addressed.

1. In the overall experiment design, mice were immunized twice with Ag and adjuvant LTA1 and then infected with the bacteria for challenge. After a 24 hr challenge, the adaptive immune cells including Th17 cells were investigated. However, the bacterial challenge could induce changes of the cell population. Thus, I am not sure whether each deficient condition such as IL-1R KO influences the generation of Th17 by vaccination or Th17 TRM reactivation upon challenge. Please investigate the adaptive cells after only vaccination before challenge.

2. Genetic mice which are used in this manuscript are all global knockout mice. However, IL-1 signaling is important in immune response including both innate and adaptive response. Thus, it would be hard to exclude the effect of IL-1 deficiency or caspase 1 deficiency on the other cells to induce the immune response by vaccination. Please at least discuss cell intrinsic versus cell extrinsic effect of IL-1.

3. In the line 287 of discussion, the sentence may be over-estimated that the signaling through IL-1R plays a major role in vaccine-mediated lung Th17 TRM generation. Although IL-1R KO showed the decrease of Th17 by vaccination and challenge, it was not completely and only partial effect. Also, the author suggested the Th17 cells by vaccination and challenge are resident memory cells. However, it is not clear in this model whether the Th17 cells by vaccination and challenge are resident or not. Although there is a reference which they reported about Th17 TRM, the experimental scheme to check Th17 was different in this manuscript. In the reference, Th17 TRM was identified from the lungs before challenge by the scRNA seq analysis and also confirmed by intravenously labeling. Please revise the sentence and discuss the potential contribution of circulating Th17 in contributing to the protection.

4. The IL-1R expression levels were shown only in gene levels by sequencing analysis. The authors should validate IL-1R expression levels by RT-PCR or staining.

5. Figure 1D – IL-17A in BAL is not significant in the figure while it is mentioned as significant in the manuscripts.

6. In Fig 1 and in line 82, the title is whether vaccination impacted IL-1 related signaling in vivo. However, the results showed only inflammatory response, and seems not to relate with the IL-1 signaling. And there is no mention of the results of IL-27, IL-10 and IL-12p40. Also, there are only IL-1a and IL-1b results, not about other IL-1 cytokine families such as IL-18, IL-33, IL-37 or IL-38. Please revise the title.

7. Fig 2a, FACS dot plot makes it confusing to show high frequencies of CD4 T cells and B cells in PBS groups although the number is very low in Fig B and C. Please provide a different gating analysis from the total lung.

8. IL-1b can be cleaved in the caspase-1 independent manner, even though caspase-1 dependent

cleavage of pro-IL-1b is the major pathway. In line 150-151, please revise.

9. In line 171, there is no direct evidence about cleaved IL-1b in the effect of the vaccination, 'caspase-1 dependent IL-1b' might be a more accurate expression. Please revise.

10. In line 200-202, please add reference.

Reviewer #3 (Remarks to the Author):

Using a combination of genetic and antibody neutralization approaches, the paper proved that Th17 TRM cells can be generated independent of caspase-1 but are compromised when IL-1 α was neutralized. Moreover IL-1 α could serve as a molecular adjuvant to generate lung Th17 TRM cells independent of LTA1. Then the researchers make a point that IL-1 α plays a major role in vaccine-mediated lung Th17 TRM cells generation.

The research has a good novelty and will be of interest to others in the community and the wider field.

However, some aspects should be improved as outlined below:

1. Lines 31-32, "Taken together, these data suggest that IL-1 α plays a major role in vaccine-mediated lung TRM generation." should be "Taken together, these data suggest that IL-1 α plays a major role in vaccine-mediated lung Th17 TRM generation."
2. Lines 325-331, discussion about that previous studies' data are contradictory to the paper's results is not sufficient. More references should be mentioned and explained here such as the paper (PLoS Pathog. 2016 May 5;12(5):e1005589. doi: 10.1371/journal.ppat.1005589. eCollection 2016 May).
3. The statistical symbols such as *, **, *** are not consistent between the manuscript (line 533) and the figures.
4. Figure 2A, 3A, 4A: the values of subset cells are not clear in representative flow plots. What's more, the subset cell gated should be CD19-B220+. Please confirmed the conditions.
5. The differences of the results about the OmpX Serum IgG between Figure 4F and Figure 5C is obvious. Why?
6. In Figure 5C it is weird that there is no value for OmpX Serum IgG when the serum is diluted to 128 times.
7. Figure 4F and Figure 5: "IL-1a" and "IL-1b" should be corrected.

Article Title: “Vaccine-elicited IL-1R signaling results in Th17 TRM-mediated immunity”

Re: Response to reviewers

We thank the reviewers for the time and effort given to evaluating our manuscript. We have modified the manuscript in response to these comments, as detailed below and in the manuscript where changes to text are marked in red. Based on the provided comments, we have clarified several issues and performed additional experiments, primarily to verify Th17 cell elicitation in the absence of challenge. Overall, we believe these changes have improved the clarity of the work and strengthened our main conclusions.

Reviewer #1 (Remarks to the Author):

The manuscript by Hoffman et al. examines the contribution of IL-1alpha and IL-1beta in the generation of protective immunity through the vaccine-elicited activation of lung tissue resident memory (Trm) cells producing IL-17A. Their work implicates that IL-1alpha is both necessary and sufficient to elicit robust vaccine responses in their model system.

The studies conducted are appropriate based on widely-available tools, conveyed clearly, and presented in a straightforward manner. There are a few comments and questions that the authors could address to improve the manuscript for readers.

1. In the Introduction, describe the immunization method/site used in prior and current studies. Later reading indicates the immunization is via oropharyngeal aspiration to the lung, but it would be valuable to know this earlier. Also, consider including this information in the first paragraph of the Results.

We agree with the reviewer this information is important to include up front and have modified the introduction in line 62 and the Results in line 74 to clarify oropharyngeal instillation was used for immunizations.

2. Lines 80-81 allude to low *Il1r1* expression on naïve lung resident CD4+ T cells, but the data in Figure 1B are a comparison of lung (activated) and spleen (naïve) CD4+ T cells.

We have modified the language in line 83 to emphasize the naïve cells are isolated from the spleen due to the scarcity of naïve lung CD4+ TRM in unvaccinated SPF mice. We also included a sentence to clarify why these cells were chosen in lines 76-78: "Comparison to naïve splenic T cells was chosen due to the lack of lung resident CD4+ T cells in unvaccinated SPF mice."

3. As another inflammasome-regulated cytokine, was IL-18 measured in the samples used for Figure 1C-N?

The reviewer makes an excellent point about evaluating other inflammasome-regulated cytokines. We agree, these other cytokines could also play a role in T cell generation following administration of LTA1-adjuvanted vaccines. We focused Figure 1D-O on IL-1 α and IL-1 β , given the increased expression of IL-1R1 in vaccine-induced CD4+ T cells (Figure 1A-C), but future examination of additional pathways involved could also include other cytokines such as IL-18. Our single cell RNA sequencing data indicates that both naïve spleen cells and vaccinated lung cells express the receptor *Il18r1*, and we find that mice deficient in caspase-1, which cleaves pro-IL-18 into its active form, do not show a deficiency in production of vaccine-generated Th17 cells. However, non-canonical pathways for IL-18 activation could be impacting these results, and thus we agree a deeper look into the role of this cytokine could be warranted in future studies. We have modified the discussion in lines 402-406 to include IL-18 in evaluation of additional pathways promoting vaccine-induced Th17 cell differentiation.

4. Line 17 contains an extra "role".

The extra “role” has been removed.

5. It seems unlikely the IL-1alpha ELISA would be able to discriminate processed from unprocessed IL-1a in the lung tissue, whereas it could be assumed that the IL-1a in the serum is the processed form. Perhaps the local IL-1a in the lung comes about as a consequence of processing-independent release from damaged epithelial cells (wherein it is described as a chromatin-associated nuclear protein). This could be discussed.

The reviewer makes an excellent point about potential sources of the IL-1 α detected by ELISA in lung homogenates. Given previous studies demonstrating pro-IL-1 α has similar bioactivity to cleaved IL-1 α , it is also possible infection with *K. pneumoniae* induces lung injury and subsequent release of pro-IL-1 α , leading to the similarities we see between wild-type and *Il1a* ^{Δ 559,1} mice. We have included language in the text, lines 256-263, to discuss these possibilities.

6. Are Trm cells increased in the IL-1a-adjuvanted immunization protocol and are these cells decreased in the mice in which IL-1a (or IL-1b) have been inhibited?

We thank the reviewer for the excellent question. We performed IL-17A ELISpot on lung cells isolated from IL-1 α -adjuvanted animals prior to challenge to determine the number of vaccine-induced Th17 cells. We found that Th17 cells increased in mice post-vaccination, and when the same experiment was performed in IL1R1KO mice (where IL-1-mediated signaling is abrogated), we found a lower number of generated Th17 cells (included now as Supplementary Figure 5). We suspect the majority of these vaccine-induced Th17 cells are TRMs based on previous data demonstrating that ~90% of Th17 cells isolated from lungs of vaccinated mice were tissue resident (PMID: 34516780).

Supplementary Figure 5: IL-1 signaling-mediated generation of OmpX-specific lung resident Th17 cells is dependent on OmpX + IL-1 α immunization. ELISpot results measuring IL-17A secreting cells after overnight stimulation with OmpX. Data are displayed as spot counts per 10⁵ plated cells from lung homogenate. Data are represented as mean \pm SEM, n = 3-4 mice per group. Statistical differences were determined using a one-way ANOVA. **, p < 0.01; ***, p < 0.001.

7. The authors could mention the potential of using the newly-described IL-1alpha-deficient mice generated by Dr. Kanneganti's group (PMC9729281) to provide additional insight into the contribution of IL-1a in this vaccine model.

We thank the reviewer for the suggestion and have added reference to these mice in the discussion in lines 354-356: "Importantly, these studies could benefit from a more recently developed IL-1 α -deficient mouse model, which does not affect IL-1 β and could potentially better illustrate the role of endogenous IL-1 α in vaccine-mediated Th17 cell generation"

Reviewer #2 (Remarks to the Author):

The authors, Hoffmann and colleagues, described that immunization of OmpX from K. pneumoniae with adjuvant LTA1 induces Th17 mediated by IL-1 signaling. They proposed the underlying mechanism by which the mixture of adjuvant LTA1 and antigen OmpX elicit antigen specific lung Th17 cells and further suggested the role of IL-1a as an adjuvant. Overall, this is well studied about the importance of IL-1 in the Th17 induced by vaccination. However, there are several points which need to be addressed.

1. In the overall experiment design, mice were immunized twice with Ag and adjuvant LTA1 and then infected with the bacteria for challenge. After a 24 hr challenge, the adaptive immune cells including Th17 cells were investigated. However, the bacterial challenge could induce changes of the cell population. Thus, I am not sure whether each deficient condition such as IL-1R KO influences the generation of Th17 by vaccination or Th17 TRM reactivation upon challenge. Please investigate the adaptive cells after only vaccination before challenge.

To address this point, we performed flow cytometry and ELISpot analysis of lung cells from unchallenged animals that had been immunized with LTA1+OmpX to determine the number of vaccine-induced Th17 cells in the absence of challenge. We found that Th17 cells increased in mice post-vaccination, and when the same experiment was performed in IL1R1KO mice, we found a lower number of generated Th17 cells (new Supplementary Figure 2 and below). These data demonstrate that vaccination is influencing the generation of Th17 independent of bacterial challenge.

Supplementary Figure 2: IL-1 signaling-mediated generation of lung resident Th17 cells is dependent on OmpX + LTA1 immunization. Quantification of (A) CD4 T cells, (B) B cells, and (C) Th17 cells in the lungs of vaccinated and unvaccinated wild-type and *Il1r1*^{-/-} mice as measured in flow cytometry. (D) Representative flow plots for determining lung Th17 cells (CD3⁺, CD4⁺, TCR-β⁺, IL-17A⁺) in immunized and unimmunized wildtype and *Il1r1*^{-/-} mice. (E) ELISpot results measuring IL-17A secreting cells after overnight stimulation with OmpX. Data are displayed as spot counts per 10⁵ plated cells from lung homogenate. (F) Representative flow plots for determining lung TRM cells (CD3⁺, CD4⁺, TCR-β⁺, CD44⁺, CD69⁺) in unimmunized wildtype and immunized wildtype and *Il1r1*^{-/-} mice. (G) Quantification of the total number of lung TRM cells in the lungs of vaccinated and unvaccinated wild-type and *Il1r1*^{-/-} mice as measured by flow cytometry. Antibodies for flow were as follows: APC rat anti-mouse CD3e (clone 17A2, Biolegend), PerCP-Cy5.5 rat anti-mouse CD4 (clone GK1.5, Biolegend), PE-Cy5 hamster anti-mouse TCRβ (clone H57-597, BD Biosciences), FITC rat anti-mouse IL-17A (clone TC11-18H10.1, Biolegend), AlexaFluor 647 rat anti-mouse IFN_γ (clone XMG1.2, Biolegend), PE-Cy7 hamster anti-mouse CD69 (clone H1.2F3, eBioscience), and AlexaFluor 700 rat anti-mouse/human CD44 (clone IM7, Biolegend). Data are represented as mean ± SEM, n = 2-4 mice per group. Statistical differences were determined using a one-way ANOVA. *, p < 0.05; **, p < 0.01; ***, p < 0.001; ****, p < 0.0001.

2. Genetic mice which are used in this manuscript are all global knockout mice. However, IL-1 signaling is important in immune response including both innate and adaptive response. Thus, it would be hard to exclude the effect of IL-1 deficiency or caspase 1 deficiency on the other cells

to induce the immune response by vaccination. Please at least discuss cell intrinsic versus cell extrinsic effect of IL-1.

We recognize the issues raised with germline KOs. To complement our KO animal studies, we included data using anti-IL1 antibodies delivered oropharyngeally to investigate transient and local loss of IL-1 signaling as well as using IL-1 α alone as an adjuvant to stimulate Th17 cell production (Figures 4 and 5). We have revised the discussion starting at line 392 to highlight the issue of germline KO mice and address shortcomings in these studies.

3. In the line 287 of discussion, the sentence may be over-estimated that the signaling through IL-1R plays a major role in vaccine-mediated lung Th17 TRM generation. Although IL-1R KO showed the decrease of Th17 by vaccination and challenge, it was not completely and only partial effect. Also, the author suggested the Th17 cells by vaccination and challenge are resident memory cells. However, it is not clear in this model whether the Th17 cells by vaccination and challenge are resident or not. Although there is a reference which they reported about Th17 TRM, the experimental scheme to check Th17 was different in this manuscript. In the reference, Th17 TRM was identified from the lungs before challenge by the scRNA seq analysis and also confirmed by intravenously labeling. Please revise the sentence and discuss the potential contribution of circulating Th17 in contributing to the protection.

The reviewer makes a great point that resident cells were not clearly distinguished from circulating cells. Our previous work demonstrated that ~90% of Th17 cells isolated from lungs of vaccinated mice were tissue resident. Moreover, vaccination was maintained in the presence of FTY720, supporting that circulating T cells have a limited role in LTA1+OmpX immunization. Thus, we have revised the text starting on line 325 to read “We have demonstrated that signaling through IL-1R1 plays a contributing role in vaccine-mediated lung Th17 cell generation, the majority of which we expect to be TRM cells.” We have also added language to discuss this point starting in line 149: “We previously demonstrated that the majority (~90%) of vaccine-elicited Th17 cells in the lungs were tissue resident, suggesting IL-1R1 signaling may play a role in Th17 TRM generation.”

4. The IL-1R expression levels were shown only in gene levels by sequencing analysis. The authors should validate IL-1R expression levels by RT-PCR or staining.

We thank the reviewer for the suggestion and have included this data now as Figure 1C (and below). Using flow cytometry, we found that IL-1R1 expression was increased in CD4+ T cells following immunization, supporting the sequencing data presented in Figure 1.

(C) Expression level of IL-1R1 measured by flow cytometry in PBS treated and OmpX+LTA1 vaccinated mice.

5. Figure 1D – IL-17A in BAL is not significant in the figure while it is mentioned as significant in the manuscripts.

We thank the reviewer for the catch. We have revised that section of the manuscript in lines 95-97 to read “we saw a significant increase in IL-17A in the lung homogenate, and a small but significant increase in the serum (Figure 1E). The BALF values for IL-17A were not significant ($p = 0.054$) but demonstrated a trend that matched the other samples tested.”

6. In Fig 1 and in line 82, the title is whether vaccination impacted IL-1 related signaling *in vivo*. However, the results showed only inflammatory response, and seems not to relate with the IL-1 signaling. And there is no mention of the results of IL-27, IL-10 and IL-12p40. Also, there are only IL-1a and IL-1b results, not about other IL-1 cytokine families such as IL-18, IL-33, IL-37 or IL-38. Please revise the title.

The reviewer raises an excellent point. Based on the increase in IL-1R1 we observed in scRNAseq, we focused on ligands for this receptor, which is why other IL-1 family members were not included. Thus, we have revised the Figure 1 caption title to be “OmpX + LTA1 immunization promotes *Il1r1* transcript expression in CD4 T cells and induces IL-1 and other proinflammatory cytokines *in vivo*.” In addition, we have changed the text in the first two paragraphs of the Results to more accurately reflect that data presented are on the inflammatory response to vaccination and not looking at IL-1 signaling broadly. We have also included discussion of the results for IL-27, IL-10, IL12p70, and IFN β in lines 101-105.

7. Fig 2a, FACS dot plot makes it confusing to show high frequencies of CD4 T cells and B cells in PBS groups although the number is very low in Fig B and C. Please provide a different gating analysis from the total lung.

We thank the reviewer for this suggestion and acknowledge the original presentation was unclear, where Figure 2A showed frequency of T and B cells in CD3+ or CD3- populations, respectively, whereas Figures B and C showed frequencies in 50,000 events of all cells. We have revised figures 2, 3, 4, and S3 to remove the misleading flow plots and only report the total

number of cells identified in each population, based on the gating strategy outlined in Supplementary Figure 1.

8. IL-1b can be cleaved in the caspase-1 independent manner, even though caspase-1 dependent cleavage of pro-IL-1b is the major pathway. In line 150-151, please revise.

We thank the reviewer for pointing this out, and have revised text in lines 177-179 to read “Without caspase-1, *Casp1*^{-/-} mice are unable to generate active IL-1β via the canonical caspase-1-dependent pathway, though other pathways could contribute to activated IL-1β production .” In addition, we addressed this possibility in the discussion starting at lines 367.

9. In line 171, there is no direct evidence about cleaved IL-1b in the effect of the vaccination, ‘caspase-1 dependent IL-1b’ might be a more accurate expression. Please revise.

The reviewer makes an excellent point. We have revised text in lines 199-201 to read “These data demonstrate that caspase-1 dependent IL-1β is not required for the generation of lung Th17 cells or vaccine-mediated protection in our model.”

10. In line 200-202, please add reference.

We have added a reference for the text in lines 228-230.

Reviewer #3 (Remarks to the Author):

Using a combination of genetic and antibody neutralization approaches, the paper proved that Th17 TRM cells can be generated independent of caspase-1 but are compromised when IL-1α was neutralized. Moreover IL-1α could serve as a molecular adjuvant to generate lung Th17 TRM cells independent of LTA1. Then the researchers make a point that IL-1α plays a major role in vaccine-mediated lung Th17 TRM cells generation.

The research has a good novelty and will be of interest to others in the community and the wider field.

However, some aspects should be improved as outlined below:

1. Lines 31-32, “Taken together, these data suggest that IL-1α plays a major role in vaccine-mediated lung TRM generation.” should be “Taken together, these data suggest that IL-1α plays a major role in vaccine-mediated lung Th17 TRM generation.”

We thank the reviewer for carefully reading the manuscript and we have revised the line to read “lung Th17 TRM.”

2. Lines 325-331, discussion about that previous studies’ data are contradictory to the paper’s results is not sufficient. More references should be mentioned and explained here such as the paper (PLoS Pathog. 2016 May 5;12(5):e1005589. doi: 10.1371/journal.ppat.1005589. eCollection 2016 May).

We have strengthened this section as suggested by the reviewer in the third paragraph of the Discussion and included the suggested reference. We have also clarified the language to make it clearer that IL-1 β is a well-established pathway through which Th17 differentiation is induced. We did not intend to suggest our data is contradictory to previous studies' data; instead, we suggest data from Figure 4 is somewhat contradictory to what we found in Supplemental Figure 3 and further supports the role of IL-1 β in Th17 cell generation. We also argue that LTA1 mediated generation of Th17 cells in the context of our mucosal *K. pneumoniae* vaccine is perhaps preferentially mediated through IL-1 α , though IL-1 β likely plays a role as well, as demonstrated in Figure 4 with antibody inhibition of IL-1 β . We have also emphasized the shortcomings in our IL-1 β analysis, including that we did not account for all non-canonical pathways of IL-1 β processing and that global knockout mice could be affecting additional immune signaling pathways that play a role in Th17 cell generation.

3. The statistical symbols such as *, **, *** are not consistent between the manuscript (line 533) and the figures.

We thank the reviewer for catching this error and have modified the text to make the values of the symbols consistent between what is stated in the methods and what is presented in the figure captions.

4. Figure 2A, 3A, 4A: the values of subset cells are not clear in representative flow plots. What's more, the subset cell gated should be CD19-B220+. Please confirmed the conditions.

We thank the reviewer for the comments and acknowledge that our original representative flow plots were unclear. We have revised all of the representative flow plots (Figures 2, 3, 4, S3) to more clearly report the gated population. Further we have amended the figures to report the B cells as CD3-B220+, omitting the CD19 gating as staining with this fluorophore was not consistent. The gating strategy has been updated in Supplementary Figure 1.

5. The differences of the results about the OmpX Serum IgG between Figure 4F and Figure 5C is obvious. Why?

We expect this difference is likely due to the differences in the vaccination protocol. In Figure 4, we used 10 μ g LTA1 admixed with 1 μ g OmpX for immunizations, whereas in Figure 5, we used 2.5 μ g IL-1 α or IL-1 β admixed with 1 μ g OmpX. We believe the change in adjuvant dose likely contributed to differences in the serum antibody titers to OmpX we see when comparing the two different vaccines. Further studies would be needed to optimize the vaccine formulation using the IL-1 α or IL-1 β adjuvants which could improve titers of anti-OmpX serum IgG elicited.

6. In Figure 5C it is weird that there is no value for OmpX Serum IgG when the serum is diluted to 128 times.

We agree with the reviewer, the graph is misleading. As 1/128 and 1/32768 diluted serum were not tested, these values should not be included on the x axis. We have modified the x axis to remove these points and only go from 256-16384, the full range of dilutions tested.

7. Figure 4F and Figure 5: “IL-1a” and “IL-1b” should be corrected.

We thank the reviewer for catching these errors and have made the corrections to Figures 4F and 5.

REVIEWERS' COMMENTS:

Reviewer #1 (Remarks to the Author):

The authors have substantially modified the manuscript in response to the recommendations of all three reviewers. These changes have improved the manuscript overall.

Reviewer #2 (Remarks to the Author):

The original manuscript by Hoffman et al. described the contribution of IL-1R1 signaling in vaccine induced Th17 TRM cells. The revised manuscript addresses the comments from reviewers; however, it still contains a few minors that need to be fixed.

1. Line 130-131, "IL-1b signaling does not have an impact on the enrichment of total adaptive immune populations in our vaccine model". However, the figures do not total adaptive immune populations, only CD4 and B cell data are shown. Please revise this.
2. Line 313-314, "differences between PBS administered and rIL-1a adjuvanted groups were not significant". In Figure 5B, there are marked as significant. Please correct the manuscript or figure.
3. In the legends of Figure 2, 3, and 4, 'CD4+, TCR-β+' are repeated. Please delete one of them.
4. Figure 4D, the labels and dot plots of 'Naïve' and 'Isotype' do not seem to match. Please confirm it.
5. Supplementary Figure 1, the label (SSC-A) of the third dot plot is not correct. Please confirm it.
6. Supplementary Figure 2, in panel G, there is omitted in the figure. Please mark 'G' on the figure.

Reviewer #3 (Remarks to the Author):

The revised manuscript has addressed my concerns. I have no more questions up to now. Thanks a lot.

Article Title: “Vaccine-elicited IL-1R signaling results in Th17 TRM-mediated immunity”

Re: Response to reviewers

We thank the reviewers again for evaluating our manuscript. We have made the suggested modifications to the manuscript, and responses to reviewer comments can be found below. We have also corrected a minor error found in the data entry for Supplementary Figure 2a, which did not change the overall findings or conclusions of the paper. Modifications to the text describing this figure can be found in lines 140-144. Changes made to text in the manuscript are marked in red.

Reviewer Comments:

1. Line 130-131, “IL-1b signaling does not have an impact on the enrichment of total adaptive immune populations in our vaccine model”. However, the figures do not total adaptive immune populations, only CD4 and B cell data are shown. Please revise this.

We agree with the reviewer, since total adaptive immune populations were not examined, we have revised the text to be more accurate as follows: “Given these data, it appears that IL-1 signaling does not have an impact on the enrichment of B and CD4+ T cell populations in our vaccine model.”

2. Line 313-314, “differences between PBS administered and rIL-1a adjuvanted groups were not significant”. In Figure 5B, there are marked as significant. Please correct the manuscript or figure.

We appreciate the catch by the reviewer. The differences between the rIL-1a adjuvanted and PBS groups are significant, and thus we have revised these lines to read: “Additionally, 3 out of 7 of the IL-1 α adjuvanted mice were protected from bacterial dissemination as evidenced by having no detectable bacteria in the spleen, while the remaining 4 mice had similar burdens to PBS treated mice. However, differences between PBS administered and rIL-1 α adjuvanted groups remained significant (P = 0.0262, Figure 5b).”

3. In the legends of Figure 2, 3, and 4, ‘CD4+, TCR- β +’ are repeated. Please delete one of them.

We appreciate the catch and have revised each caption accordingly.

4. Figure 4D, the labels and dot plots of ‘Naïve’ and ‘Isotype’ do not seem to match. Please confirm it.

We had incorrectly labeled naïve and isotype dot plots and have switched the labels in the figures.

5. Supplementary Figure 1, the label (SSC-A) of the third dot plot is not correct. Please confirm it.

The reviewer is correct, the y-axis label should be “B220”. We have modified the figure accordingly.

6. Supplementary Figure 2, in panel G, there is omitted in the figure. Please mark 'G' on the figure.

We have added 'g' to Supplementary Figure 2g to match the caption.